# How Servant Leadership Motivates Innovative Behavior: A Moderated Mediation Model

**DOI:** 10.3390/ijerph17134753

**Published:** 2020-07-02

**Authors:** Jianji Zeng, Guangyi Xu

**Affiliations:** 1School of Medical Business, Guangdong Pharmaceutical University, Guangzhou 510006, China; zengjianji2006@163.com; 2School of Business Administration, South China University of Technology, Guangzhou 510640, China

**Keywords:** servant leadership, perceived insider status, organization-based self-esteem, innovative behavior, LMX

## Abstract

Drawing on social identity theory, this study examines the effect of servant leadership on university teachers’ innovative behavior through the self-concept constructs of perceived insider status and organization-based self-esteem, and the moderating effect of leader–member exchange (LMX). This moderated mediation model was tested with two waves of data from 269 university teachers in China. Results reveal that the self-concept constructs mediate the relationship between servant leadership and university teachers’ innovative behavior. Moreover, LMX strengthens the relationship between servant leadership and the self-concept constructs, as well as the indirect effect of servant leadership on university teachers’ innovative behavior through the self-concept constructs. Findings suggest that servant leadership is related to increased innovative behavior due to its positive influence on the self-concept of university teachers and it highlights the importance of developing a favorable supervisor–subordinate relationship.

## 1. Introduction

Workplace innovation has been a hot topic among organizational scientists over the last few decades [1]. Innovation is generally viewed as a key factor in the success of organizations, especially in knowledge-based societies [2,3]. Organizational innovation stems from employee innovative behavior [4], which is described as self-initiated behavior to generate and implement new ideas to benefit the individual or organization [5]. Given the importance of innovative behavior, an increasing number of studies explore how to motivate employee innovative behavior [6].

The extant literature suggests leadership is a critical predictor of employee innovative behavior [7,8,9]. Numerous leadership constructs have been shown as antecedents of workplace innovation [8]. Given the conceptual issues associated with established leadership constructs [10,11], the examination of contemporary leadership styles (e.g., servant leadership) has increased recently [8]. The altruism in servant leadership has been proven to have positive influences on employee innovation or creativity [12,13]. Servant leaders can create a climate of trust, which motivates employees to take risks and develop innovative ways [14]. In a meta-analysis, Lee et al. [15] suggested that servant leadership is positively related to individual creativity.

The majority of the literature about workplace innovation focuses on highly competitive situations wherein organizations and employees need to engage in innovative behavior for the survival and sustainability of these organizations. In fact, not only organizations in competitive environments need to innovate, but nonprofit organizations as well, such as educational institutes [3]. Universities are the core force of social innovation systems, and university teachers play an important role in cultivating innovative talents. Several reasons explain why university teachers’ innovative behavior is needed. First, university teachers’ engagement in innovative behavior is important to keep pace with a rapidly changing society. Innovative behavior is beneficial to update teaching technologies and learn new insights. Furthermore, given that education is crucial to cultivate students’ innovative thinking [16], universities need to set a good example for innovation so that society can maintain competitiveness; that is, university teachers’ innovative behavior is highly important for the development of knowledge-based societies and educational professions [3].

This study aims to examine how servant leadership affects university teachers’ innovative behavior. Following prior studies [13,17], we propose that servant leadership is positively related to university teachers’ innovative behavior. University teachers will engage in innovative behavior to reciprocate servant leadership. Given the characteristics of servant leaders [18], we expect that the potential benefits of servant leadership behaviors also hold for university teachers. Moreover, drawing on the cultural self-representation model [19], we examine the self-concept constructs of perceived insider status (PIS) and organization-based self-esteem (OBSE) as potential mediators in the servant leadership–innovative behavior relationship. Given that the effectiveness of servant leadership will be influenced by the leader–follower relationship [20], we further investigate the moderating effect of the relational factor of leader–member exchange (LMX) on the relationship between servant leadership and the self-concept constructs.

The contribution of this study involves the following aspects. First, drawing on self-concept theory [21], we emphasize that the followers’ self-concept embedded in a group, which includes PIS and OBSE, transmits the effect of servant leadership on innovative behavior. This effort extends previous studies that have shown mediating influences played by motivational [22,23,24], cognitive [25,26], and identification-based mechanisms [13]. Secondly, the boundary conditions about servant leadership largely focus on employee-centered moderators [23,27,28] or organization-centered moderators [29,30,31]. This study focuses on LMX to shed light on the leader–follower relationship as a potential moderator. Therefore, we construct a moderated mediation framework with the self-concept constructs as the mediators and LMX as the moderator. Finally, the majority of the innovative behavior literature focused on business organization, thus limiting the generalizability of the findings. Non-profit organizations refer to those social organizations that are non-profit-oriented, whose mission is to solve social problems and create social values. According to the International Classification of Non-Profit Organizations, non-profit organizations are divided into 12 major groups and 27 subgroups, and higher education belongs to the group of education and research. The core functions of universities include personnel training, scientific research, and social services; universities are generally regarded as a special non-profit organization. Using a sample of university teachers in China, this study enriches the understanding of innovative behavior in non-profit organizations.

## 2. Theoretical Background and Hypotheses

To understand the potential benefits of servant leadership behaviors for university teachers, we draw on social identity theory [32], which is used to explain how servant leaders make employees feel like they are part of an organization [33,34]. This theory focuses on how individuals enhance their positive self-image and self-esteem by relating their self-concept to other social groups [35,36]. Social identity theorists suggest that social identity enables an individual to achieve a positive self-concept [35,36,37].

According to social identity theory, university teachers will achieve a positive self-concept when they are treated by their leaders with respect, politeness, and acceptance. The interpersonal treatment from supervisors is usually regarded as a cue for employees to make inferences about their status and value [38,39]. Furthermore, we posit that university teachers with a positive self-concept are likely to engage in innovative behavior, which is beneficial to themselves and their organization [34]. Linking servant leadership literature with social identity theory, we propose that servant leaders facilitate university teachers’ innovative behavior due to its positive influence on the self-concept of university teachers, indicated by their PIS and OBSE.

### 2.1. Servant Leadership and Innovative Behavior

The term servant leadership was coined by Greenleaf [40], who stated that “Servant–Leader is servant first. It begins with the natural feeling that one wants to serve first”. Although this statement is the most well-known description of servant leadership, it is inadequate for guiding empirical study [33]. Eva et al. [33] defined servant leadership as an other-oriented leadership, in which leaders give priority to their followers’ individual interests and needs, and is reoriented toward concerns for others within an organization and the larger community. Van Dierendonck [18] presented the characteristics of servant leadership, such as empowering and developing people, humility, authenticity, interpersonal acceptance, providing direction, and stewardship. In line with the “acid test” of servant leadership [40], employees under servant leadership are likely to engage in positive work-related behaviors, such as organizational citizenship behavior, proactive behavior, and innovative behavior [33].

Innovative behavior refers to the production and implementation of useful ideas [41,42], which may entail considerable risk taking [43]. Numerous studies suggested that leadership plays a critical role in the process of innovation [44,45]. Krause [46] developed a model to explain how leadership affects the cognitive process of perceiving that the work setting needs to change and developing innovative behaviors (generate and implement new ideas). The altruism in servant leadership has been proven effective to promote individual creativity or innovative behavior [15].

According to social identity theory, social identity is the perception of membership to a social group. Specific emotions and values are created because of this perception of membership to a group [47,48], and individuals tend to select and implement activities that are consistent with their social identity [37]. As an other-oriented leadership, servant leaders are willing to empower and provide opportunities to followers. Owing to the dyadic relationship with servant leaders, employees are likely to develop a strong sense of belonging and acceptance [17]. Namely, employees under servant leadership are more likely to develop a positive self-concept, which will motivate them to engage in innovative behavior. Specifically, servant leaders establish close bonds with their followers, who are likely to perceive themselves as insiders, thus developing an intrinsic motivation to engage in innovative behavior [17]. Accordingly, servant leaders make followers feel emotionally safe and hence increase their willingness to generate new ideas and initiate change. Moreover, the purpose of empowering people is facilitating a proactive attitude among followers and developing a sense of personal power [18], which makes employees feel autonomous and take new challenges. Previous studies also provided support for the positive influence of servant leadership on employee innovative behavior [12,13,17]. Thus, we hypothesize the following:

**Hypothesis** **1** **(H1).**
*Servant leadership is positively related to university teachers’ innovative behavior.*


### 2.2. The Mediating Role of Self-Concept

Although social identity theory provides a potential perspective for understanding the servant leadership–innovative behavior relationship, the meditating mechanism needs to be further explored. Given that servant leadership reflects high-quality organization–employee relationships, it enables employees to incorporate their statuses as organization members into their self-concept. The influence of servant leadership on innovative behavior can be explained with regard to the experience of self-direction that servant leadership engenders. This experience will motivate employees to challenge themselves and try new ways of working, thus contributing to innovative behavior [43].

Accordingly, we expect that self-concept is an important link in the servant leadership–innovative behavior relationship. Self-concept refers to the totality of an individual’s thoughts and feelings in relation to themselves [49], which involves the dimensions of self-conception and self-evaluation [50]. In an organizational context, self-concept can be disguised as PIS and OBSE, which reflect the self-conception and self-evaluation dimensions of the self-concept, respectively [43]. PIS refers to the extent to which employees perceive themselves to be organizational insiders [51], which reflects a sense of having earned a “personal space” and acceptance inside in their organization [52]. In contrast, OBSE refers to the degree to which organizational members believe that they can satisfy their role’s needs [53], which reflects the self-perceived value of importance, competence, and capability within their employing organizations [54]. The potential mediating mechanism of the two self-concept constructs is predicted based on the notion that the effect of servant leadership on the followers’ innovative behavior is due to the motivational implications of the self-concept [55,56].

Following social identity theory [48], we expect servant leadership to be associated with the self-concept constructs of PIS and OBSE. For one thing, Takeuchi et al. [38] suggested that a follower’s sense of belonging to a group is influenced by the interpersonal treatment from leaders. Servant leaders establish close connections with followers through their follower-centric nature, which makes employees perceive themselves to be partners in the organization [33]. Thus, employees likely perceive themselves as in-group rather than out-group members [57]. Accordingly, Liden et al. [25] suggested that servant leadership gives priority to followers’ individual interests and needs, which helps followers develop an identity interlinked with the work group represented by the leaders. Furthermore, servant leadership is associated with PIS because being empowered to take responsibility for certain activities signals an individual’s respected position within an organization. For another, employees with high OBSE are likely to believe that “I count around here” [54]. As an other-oriented leader behavior, servant leadership will facilitate high levels of OBSE because servant leaders emphasize empowering and developing people. Such empowerment facilitates a proactive work attitude and a sense of ability among followers [18]. This perception signals to followers that supervisors or organizations consider them important, task-competent, and need-satisfying within the organization [54,58]. Further, servant leaders focus on the development of employees’ skills, competence, and abilities [59], and they provide direction for employees to clarify what is expected of them [18]. The incorporation of such positive support into an employee’s self-concept can enhance OBSE.

Additionally, we expect the self-concept constructs of PIS and OBSE to be related to innovative behavior. According to social identity theory, employees with a positive self-concept are likely to engage in activities that are consistent with and beneficial to their group membership [34]. On the one hand, the influence of PIS on innovative behavior can be explained by the motivational implications of the membership. Stamper and Masterson [51] argued that individuals who perceive themselves to be organizational insiders are likely to accept the responsibilities of citizenship. This acceptance is consistent with social identity theory, which explains why organizational members are willing to undertake extra-role activities that are beneficial to their own and their organization’s future well-being [48]. The sense of belonging along with PIS will motivate employees to perform the prescribed work role and engage in discretionary work roles, such as innovative behavior [43]. The existing literature suggested that PIS can stimulate employee innovative behavior [17,43,60]. On the other hand, self-consistency may explain the influence of OBSE on innovative behavior [54]. Social identity theory suggests that individuals with a positive social identity will strive to maintain and enhance their self-esteem [38]. Korman [61] argued that individuals with high OBSE are willing to engage in behaviors that strengthen their positive self-cognition, even if innovative behavior is often accompanied by a fair amount of risks [41]. Employees who feel capable and competent may take risks and engage more in innovative behavior [43]. Therefore, we expect the effect of servant leadership on innovative behavior to be indirect, operating through the motivational implications of PIS and OBSE. Thus, we hypothesize the following:

**Hypothesis** **2** **(H2).**
*PIS and OBSE both mediate the servant leadership–innovative behavior relationship.*


### 2.3. The Moderating Role of LMX

Positive interpersonal relationships between leaders and followers are a key factor for servant leadership to play its role [20]. Hence, we examined LMX as a moderator in the servant leadership–self-concept relationship. LMX refers to the quality of exchange relationship between leaders and followers based on trust, respect, and obligation [62]. According to LMX theory, high- to low-quality supervisor–subordinate relationships will be formed, and the quality of the LMX may be reflected in the followers’ self-concept; that is, individuals who develop high-quality relationships with their leaders will be attached psychologically to their work group [63], paving the way for social identification and fostering the perceptions of insider status and self-esteem [17]. 

The core of servant leadership is to believe in the intrinsic value of each individual [64]. Servant leaders who show humility, authenticity, and interpersonal acceptance create a favorable working environment, where followers feel trustworthy [18]. Therefore, the effect of servant leadership on followers’ self-concept may be influenced by the quality of the LMX. According to social identity theory, relational factors, such interpersonal interaction, may affect how individuals identify with a group [38]. Tajfel and Turner [48] argued that social identity will be more influential when individuals establish a strong emotional connection with the group; that is, individuals with high-quality LMX may perceive themselves as in-group members, and they can achieve more valuable resources, such as benefits, training, and promotion, which will signal to employees that they have gained insider status [43]. Meanwhile, self-esteem is partly rooted in the valuable messages transmitted from an organization to its employees [61,65]. Such individuals may develop a sense of trust and be more confident to carry out work-related activities; that is, servant leadership will be instrumental for individuals with a high-quality LMX in enhancing not only their perception of their status in a group (PIS) but also their belief that “I count around here” (OBSE).

In contrast, a low-quality LMX is based on economic exchange, which is characterized by low levels of trust, respect, and infrequent interactions between leaders and followers. Employees with low-quality LMX will perform their duties and roles according to the employment contract [62]. Accordingly, we can expect that the positive effect of servant leadership on the self-concept constructs of PIS and OBSE will be less obvious for individuals with low-quality LMX. Individuals with low-quality LMX are likely to perceive themselves as out-group member [62,66]. Further, they feel unable to gain the necessary resources, and their contribution is hardly recognized by their leaders [67], which will lead to low employee self-esteem. We can expect that a low quality LMX will weaken the positive effect of servant leadership on the self-concept constructs.

**Hypothesis** **3** **(H3).**
*LMX moderates the relationships between servant leadership and both PIS and OBSE in such a way that the relationships will be stronger for individuals with high-quality LMX than those with low-quality LMX.*


Based on the hypothesis above, we further propose a moderated mediation model (see Figure 1); that is, the quality of the LMX moderates the indirect effect of servant leadership on innovative behavior via the self-concept constructs of PIS and OBSE. Namely, the indirect effect of servant leadership on innovative behavior via the self-concept constructs will be stronger for individuals with high-quality LMX. Thus, we propose the following:

**Hypothesis** **4** **(H4).**
*LMX moderates the indirect effect of servant leadership on innovative behavior via both PIS and OBSE in such a way that the indirect effect is stronger for individuals with high-quality LMX than those with low-quality LMX.*


## 3. Method

### 3.1. Participants and Procedure

A cross-areas survey was conducted among university teachers in China in various provinces (e.g., Guangdong, Hunan, and Sichuan). We recruited survey participants though acquaintance social networks. We explained our research purpose and the anonymity of the survey, and we collected the data though the “Questionnaire Star” (a professional online questionnaire platform). The time-lagged design was used to reduce the potential common method bias. Specifically, the servant leadership, LMX, and demographic variables were assessed at Time 1. Two weeks later, PIS, OBSE, and innovative behavior were measured at Time 2. In total, 300 questionnaires were distributed via a survey link to participants, and we received 269 valid questionnaires.

Among the valid samples, 48.7% were female and 51.3% were male. Among them, 9.7% were 30 years old or younger, 39.8% were 31–40, 40.5% were 41–50, and 10.0% were 51 or older. In terms of education, 12.6% had a bachelor’s degree, 46.5% had a master’s degree, and 40.9% had a doctoral degree. Regarding academic title, 47.2% were lecturers, 38.7% were associate professors, and 14.1% were professors. Regarding work experience, 39.8% had within 10 years’ experience, 46.9% had 11–20 years, and 13.4% had more than 20 years. In addition, 48.4% were from key universities and 51.6% were from average universities.

### 3.2. Measures

All measurement items were selected from established scales. We made minor modifications to ensure that all items that are applicable to our research context. All variables were measured on a seven-point Likert scale, except for the demographic variables, ranging from 1 (‘‘strongly disagree’’) to 7 (‘‘strongly agree’’).

#### 3.2.1. Servant Leadership

The 15-item scale validated by Sun and Wang [68] based on the earlier work of Barbuto and Wheeler [69] was used to measure servant leadership. A sample item is “my leader does everything he/she can to serve me”. The Cronbach’s alpha was 0.95.

#### 3.2.2. Perceived Insider Status

This construct was assessed using Stamper and Masterson’s [51] 6-item scale. A sample item is “I feel very much a part of my work organization”. The Cronbach’s was alpha 0.90.

#### 3.2.3. Organization-Based Self-Esteem

This variable was measured on a 10-item scale developed by Pierce et al. [53]. A sample item is “I can make a difference in my work organization”. The Cronbach’s alpha was 0.83.

#### 3.2.4. Innovative Behavior

Innovative behavior was measured using Scott and Bruce’s [44] 6-item scale. A sample item is “I often generate creative ideas in my work”. The Cronbach’s alpha was 0.94. Innovative behavior items have been showed in Appendix A.

#### 3.2.5. LMX

This variable was assessed using Graen and UhI-Bien’s [62] 7-item scale. A sample item is “my supervisor recognizes my potential”. The Cronbach’s alpha was 0.88.

#### 3.2.6. Control Variables

Previous studies showed that innovative behavior was affected by social-demographic variables [44,59]. Thus, participants’ gender, age, education, and title were controlled in this study. Gender was dummy-coded as 1 for “male” and 2 for “female”. Age had four categories: 30 or under, 31–40, 41–50, 51 or older. Education had three categories: bachelor’s degree, master’s degree, and doctoral degree. Title had three categories: lecturers, associate professors, and professors.

## 4. Results 

### 4.1. Common Method Variance (CVM) 

Although a time-lagged design was used to minimize the CVM, the collected data was self-reported and single-sourced, which may result in single-source bias [70]. To assess the CVM, we conducted a Harman’s single-factor test by loading all the items of the constructs into an exploratory factor analysis [71]. The results revealed that no single factor explained more than 37.38% of the covariance among the variables, indicating that the CMV was within the acceptable range.

### 4.2. Reliability and Validity

SPSS 23.0 statistical software was used to test the reliability and validity of the data. The results in Table 1 show that the Cronbach’s alpha coefficients of the scales of servant leadership, PIS, OBSE, innovation behavior, and LMX were 0.95, 0.90, 0.83, 0.94, and 0.88, respectively. All Cronbach’s alpha coefficients were greater than 0.8, suggesting that the internal consistency of each variable is good. We examined the validity of the data through exploratory factor analysis. The KMO values of each variable were 0.92, 0.84, 0.77, 0.90, and 0.83, respectively. The cumulative variance contribution rates were 76.303%, 67.78%, 69.34%, 76.93%, and 59.19%, respectively. These results indicate that the data have good reliability and validity.

### 4.3. Descriptive Statistics

Table 1 presents the means, standard deviations, and correlations for servant leadership, PIS, OBSE, innovation behavior, and LMX. The results provide preliminary support for our theoretical predictions.

### 4.4. Hypothesis Testing

We calculated variance inflation factors to test the collinearity of the variables, and all values of the variables were less than 3. Therefore, no serious collinearity exists between the independent variables in this study. Hierarchical regression analysis and bootstrapping analysis were used to test our hypotheses. As shown in Table 2, we entered only the control variables into M1. Subsequently, the independent variable of servant leadership was added in M2. The result showed that servant leadership had a significantly positive impact on innovation behavior (β = 0.26, *p* < 0.01). Therefore, H1 was supported.

We applied hierarchical regression analysis to test the mediating role of the self-concept constructs of PIS and OBSE. According to Baron and Kenny’s [72] method for testing mediation, servant leadership was related to PIS (β = 0.48, *p* < 0.01) and OBSE (β = 0.42, *p* < 0.01), as depicted in M7 and M9 of Table 2. Furthermore, the results showed that both PIS (β = 0.18, *p* < 0.01) and OBSE (β = 0.44, *p* < 0.05) were significantly related to innovative behavior when servant leadership and demographic variables were controlled for, as shown in M5 and M6. On the basis of these results, PIS and OBSE both mediate the servant leadership–innovative behavior relationship. Therefore, H2 was supported. In this study, we proposed that LMX moderates the relationship between servant leadership and the self-concept constructs of PIS and OBSE. The results showed that the interaction between servant leadership and LMX was significantly related to PIS (β = 0.13, *p* < 0.05) and OBSE (β = 0.21, *p* < 0.01), as depicted in M8 and M10. To further describe the moderating effect of LMX, we plotted the statistically significant interaction [73] (Cohen et al., 2003). The relationship between servant leadership and PIS was stronger when LMX was above average (Mean + 1 SD) than when below average (Mean − 1 SD) (see in Figure 2). The similar conclusion about OBSE can be achieved (see in Figure 3). Therefore, H3 was fully supported.

Bootstrapping analysis was used to test the moderated mediation relationship (H4). As shown in Table 3, the indirect effect of servant leadership on innovative behavior via OBSE was not significant with a low value of LMX, with a 95% confidence interval that did contain zero. The indirect effect was significant with high value of LMX, with a 95% confidence interval that did not contain zero. Moreover, the indirect effect of servant leadership on innovative behavior via PIS was significant with low and high values of LMX, with the two 95% confidence intervals not containing zero. Furthermore, the moderating effect of LMX in the servant leadership–innovative behavior indirect relationship through PIS and OBSE was significant, with the two 95% confidence intervals not containing zero. These results suggested that LMX strengthens the indirect effect of servant leadership on innovative behavior via the self-concept constructs of PIS and OBSE. Therefore, H4 was supported.

## 5. Conclusions and Discussion

As an other-oriented leadership, servant leadership has attracted considerable attention in recent years. Drawing on social identity theory, this study examines how servant leadership affects university teachers’ innovative behavior. Servant leadership was associated with innovative behavior due to its positive effect on the self-concept constructs of PIS and OBSE. Moreover, LMX strengthened the relationship between servant leadership and the self-concept constructs, as well as the indirect effect of servant leadership on innovative behavior via the self-concept constructs.

### 5.1. Theoretical Implications

The findings of this study contribute to the relevant literature in the following aspects. First, the mediating role of the self-concept constructs of PIS and OBSE provides empirical support for the cultural self-representation model [19], which suggests that the self-concept is a critical link in the relationship between managerial practices (e.g., servant leadership) and work outcomes (e.g., innovative behavior).

Secondly, the moderating influence of LMX is consistent with Erez and Earley’s [19] argument and the following literature [74]. Interpersonal relationship may affect the effectiveness of servant leadership [20]. Individuals with a high-quality LMX are likely to perceive themselves as in-group members and achieve more valuable resources. Our findings suggested that LMX as a relational factor affects the role of servant leadership in promoting the two studied self-concept constructs.

Thirdly, although researchers had discussed the effect of servant leadership on innovative behavior, the majority of these studies focused on business organizations. However, universities are the core force of the social innovation system, and university teachers play an important role in cultivating innovative talents. University teachers who are highly educated and knowledgeable in their fields pursue self-realization [75]; thus, they are likely to engage in innovative behavior. Therefore, examining the effect of servant leadership on university teachers’ innovative behavior is valuable, and our findings contribute to the generalizability of servant leadership in a new organizational context.

### 5.2. Practical Implications

Several practical implications can be derived from research findings. First, this study confirms that servant leadership is effective to foster university teachers’ innovative behavior. Managers should focus on followers’ growth and make them understand their value to the organization, which can motivate their innovative behavior. Accordingly, organizations should recruit or cultivate leaders who demonstrate the qualities of servant leadership. Secondly, the mediating role of the self-concept constructs in this study suggests that organizations should sensitize managers to the effect of their managerial practices on the self-concept of university teachers; that is, leadership training programs should include approaches and strategies that can enhance university teachers’ self-concept. Finally, the moderating influence of LMX has implications for enhancing the effectiveness of servant leadership. To make servant leadership more effective, managers should identify relational factors that would influence the effectiveness of servant leadership. Managers should establish good interpersonal relationships with their followers by being considerate of their work or personal lives.

### 5.3. Limitations and Future Research

Several limitations in this study should be acknowledged. First, although a time-lagged design was used to minimize the CVM, cause–effect relations cannot be established for all paths in our model. A longitudinal research design can be adopted for establishing the direction of causality. Secondly, the collected data were self-reported and single-sourced, which may result in single-source bias [71]. To avoid self-report bias, information about innovative behavior can be collected from other sources, such as leaders or supervisors, in future studies. Last but not least, this study is limited to university teachers in China, which is a constraint on the generalizability of the findings. Investigating whether the results will be similar in other organizational or cultural settings is necessary.

## Figures and Tables

**Figure 1 ijerph-17-04753-f001:**
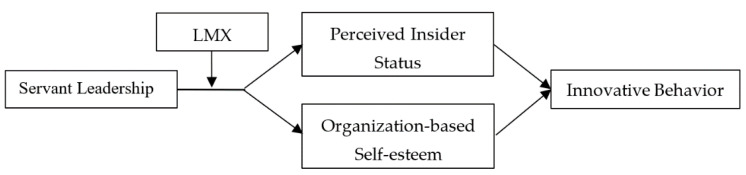
Hypothesized model.

**Figure 2 ijerph-17-04753-f002:**
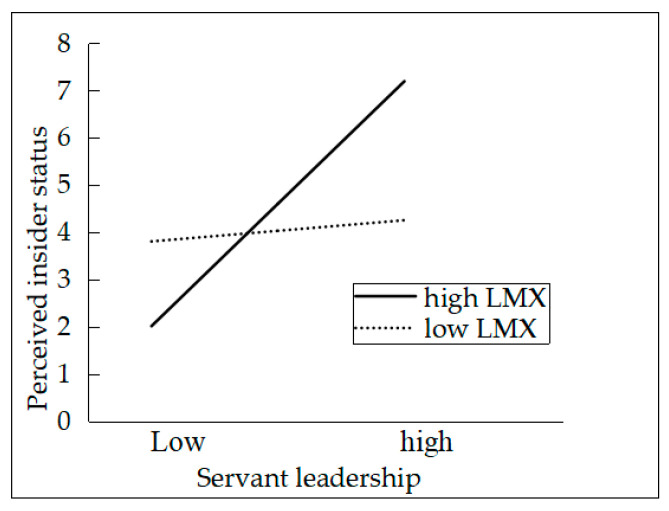
The interaction between servant leadership and LMX on PIS.

**Figure 3 ijerph-17-04753-f003:**
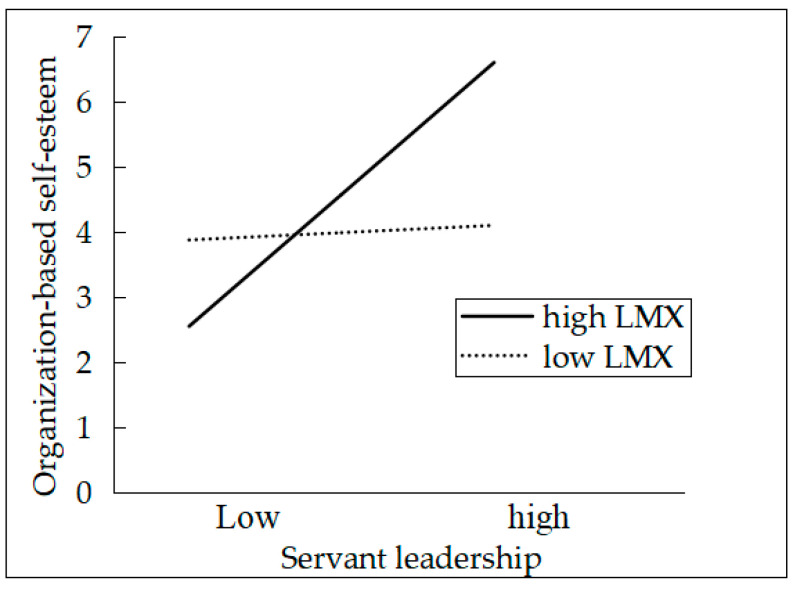
The interaction between servant leadership and LMX on OBSE.

**Table 1 ijerph-17-04753-t001:** Correlation coefficients of each variable and Cronbach’ alpha.

Variables	Mean	SD	1	2	3	4	5	6	7	8	9
1. Gender	1.49	0.50	N/A								
2. Age	40.50	1.38	−0.18 **	N/A							
3. Education	2.28	0.69	−0.07	0.05	N/A						
4. Title	1.67	0.71	−0.28 **	0.55 **	0.29 **	N/A					
5. SL	3.80	1.24	−0.04	−0.15 *	−0.12	−0.07	(0.95)				
6. PIS	4.76	1.21	−0.12 *	0.08	−0.15 *	0.06	0.47 **	(0.90)			
7. OBSE	4.67	0.85	−0.10	0.10	−0.08	0.11	0.41 **	0.52 **	(0.83)		
8. IB	5.18	0.94	−0.14 *	0.15 *	0.09	0.16 *	0.23 **	0.27 **	0.48 **	(0.94)	
9. LMX	4.09	1.09	−0.04	−0.10	−0.12	−0.04	0.79 **	0.42 **	0.46 **	0.36 **	(0.88)

Notes: N = 269; SL = servant leadership, PIS = Perceived insider status, OBSE = organization-based self-esteem, IB = innovative behavior, LMX = leader–member exchange. ** *p* < 0.01, * *p* < 0.05. Values shown in parentheses are Cronbach’ alpha of the latent variables.

**Table 2 ijerph-17-04753-t002:** Results of hierarchical regression analysis.

Variables	Dependent Variable: IB						Mediator: PIS		Mediator: OBES	
	M_1_	M_2_	M_3_	M_4_	M_5_	M_6_	M_7_	M_8_	M_9_	M_10_
Gender	−0.10	−0.08	−0.07	−0.07	−0.07	−0.06	−0.08	−0.08	−0.05	−0.05
Age	0.10	0.14 *	0.09	0.08	0.12	0.10	0.12	0.11	0.11	0.09
Education	0.06	0.09	0.11	0.11	0.12	0.12 *	−0.12 *	−0.12 *	−0.06	−0.05
Title	0.06	0.04	0.04	0.01	0.04	0.01	0.04	0.06	0.09	0.11
SL		0.26 **			0.17 *	0.07	0.48 **	0.40 **	0.42 **	0.16
PIS			0.27 **		0.18 **					
OBES				0.47 **		0.44 *				
LMX								0.10		0.34 **
SL × LMX								0.13 *		0.21 **
R^2^	0.04	0.11	0.11	0.26	0.13	0.26	0.27	0.29	0.20	0.29
ΔR^2^	−0.03	0.09 **	0.09 **	0.24 **	0.11 **	0.25 **	0.25 **	0.27 **	0.19 **	0.27 **
F	2.89 **	6.29 **	6.49 **	18.32 **	6.63 **	15.55 **	19.05 **	14.96 **	13.47 **	15.17 **

Note: N = 269; SL = servant leadership, PIS = perceived insider status, OBSE = organization-based self-esteem, IB = innovative behavior, LMX = leader–member exchange. ** *p* < 0.01, * *p* < 0.05 (two-tailed test).

**Table 3 ijerph-17-04753-t003:** Bootstrapping analysis results of the moderated mediation.

Mediators	Conditional Indirect Effects					Index of Moderated Mediation			
	Moderator	Effect	SE	95% LLCI	95% ULCI	Index	SE	95% LLCI	95% ULCI
PIS	Low LMX	0.04 **	0.02	0.01	0.10	0.01	0.01	0.01	0.04
	High LMX	0.08 **	0.03	0.03	0.14				
OBSE	Low LMX	−0.01	0.04	−0.08	0.06	0.05	0.02	0.02	0.08
	High LMX	0.10 **	0.04	0.03	0.19				

Notes: Resampling times = 5000. ** *p* < 0.01.

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
