# Peer review of "How Servant Leadership Motivates Innovative Behavior: A Moderated Mediation Model"

_ijerph, 2020, doi:10.3390/ijerph17134753_

Round 1
Reviewer 1 Report
An excellent article addressing a key research gap in servant leadership research. The literature review is solid supporting the hypotheses, and the data analysis and conclusions are sound. I have only a few minor recommendations:
- Provide more information on the sample. For example, is there data on teaching experience levels, areas of specialization, and types of universities?
- Provide in an appendix the innovation items used to facilitate a greater understanding of the dependent variable measure for the benefit of the reader.
Author Response
Response to Reviewer 1 Comments
First of all, we would like to thank the reviewer and the editor for the positive and constructive comments and suggestion.
Point 1: Provide more information on the sample. For example, is there data on teaching experience levels, areas of specialization, and types of universities?
Response 1: Thank you for your comment. A cross-areas survey on leadership and innovation behavior was conducted among 269 in Chinese universities in various provinces (e.g., Guangdong, Hunan and Sichuan). In total, 300 questionnaires were distributed via a survey link to participants, and we received 269 valid questionnaires. Among the valid sample, 48.7% are female and 51.3% are male. Among them, 9.7% are 30 years old or younger, 39.8% are 31–40, 40.5% are 41–50, 10.0% are 51 or older. In terms of education, 12.6% have a bachelor’s degree, 46.5% have a master’s degree, and 40.9% have a doctoral degree. As regards academic title, 47.2% are lecturers, 38.7% are associate professors, and 14.1% are professors. Regarding work experience, 39.8% are within 10 years, 46.9% are 11-20 years, and 13.4 % are more than 20 years. In addition, 48.4% are from key universities, 51.6% are from average universities. We have revised in lines 259-253.
Point 2: Provide in an appendix the innovation items used to facilitate a greater understanding of the dependent variable measure for the benefit of the reader.
Response 2: Thank you for your comment. We have added a appendix about innovation items in lines 399-408. Appendix: Innovation is a process involving both the generation and implementation of ideas. As such, it requires a wide variety of specific behaviors on the part of individuals. While some people might be expected to exhibit all the behaviors involved in innovation, others may exhibit only one or a few types of behavior. The innovative behavior items are as follows:
- I often searcher out new technologies, processes, techniques, and/or product ideas.
- I often generate creative ideas in my work.
- I often promote and champion ideas to others.
- I willing to investigate and secure funds needed to implement new ideas.
- I often develop adequate plans and schedules for the implementation of new ideas.
- I am innovative.

Reviewer 2 Report
The paper is well structured and organised, being understandable and easy to be read.
The topic is in accordance with the corpus of the paper and reflect the concerns of authors about the servant leadership and the impact on the innovative behaviour, tested through surveys applied on university teachers in China.
At the rows 69-71 the authors affirm „Using a sample of university teachers in China, this study enriches the understanding of innovative behavior in nonprofit organizations.” I wander if the authors considers the universities as non-profit organisations or...have something else to explain??? Moreover, at rows 49-51, the authors establish the main aim of the research paper. Maybe the authors reconsider the phrase.
Author Response
Response to Reviewer 2 Comments
First of all, we would like to thank the reviewer and the editor for the positive and constructive comments and suggestion.
Point 1: At the rows 69-71 the authors affirm, Using a sample of university teachers in China, this study enriches the understanding of innovative behavior in nonprofit organizations.” I wander if the authors considers the universities as non-profit organizations or...have something else to explain??? Moreover, at rows 49-51, the authors establish the main aim of the research paper. Maybe the authors reconsider the phrase.
Response 1: Thank you for your comment, we’ve already reconsidered the universities as non-profit organizations in lines 72-77. Non-profit organizations refer to those social organizations is profit-oriented, whose mission is to solve social problems and create social values. According to International Classification of Non-profit Organizations, non-profit organizations are divided into 12 major group and 27 subgroup, and higher education belongs to the group of education and research. The core functions of universities include personnel training, scientific research and social services, universities are generally regarded as a special non-profit organization.
